# Foliar Application of Melatonin Positively Affects the Physio-Biochemical Characteristics of Cotton (*Gossypium hirsutum* L.) under the Combined Effects of Low Temperature and Salinity Stress

**DOI:** 10.3390/plants12213730

**Published:** 2023-10-31

**Authors:** Yuanyuan Fu, Lang Xin, Abdoul Kader Mounkaila Hamani, Weihao Sun, Hongbo Wang, Abubakar Sunusi Amin, Xingpeng Wang, Anzhen Qin, Yang Gao

**Affiliations:** 1College of Water Conservancy and Architecture Engineering, Tarim University, Alar 843300, China; fyy2016060105@163.com (Y.F.); 18942777099@163.com (L.X.); 120200047@taru.edu.cn (H.W.); 13999068354@163.com (X.W.); 2Key Laboratory of Crop Water Use and Regulation, Ministry of Agriculture and Rural Affairs, Institute of Farmland Irrigation, Chinese Academy of Agriculture Sciences, Xinxiang 453002, China; sunweihao1996@163.com (W.S.); sunusiabubakar@yahoo.com (A.S.A.); 2020y90100004@caas.cn (A.Q.); 3College of Tropical Crops, Hainan University, Haikou 570100, China; 4Western Agricultural Research Center, Chinese Academy of Agricultural Sciences, Changji 831100, China

**Keywords:** cotton seedlings, ion homeostasis, foliar melatonin, membrane damage, low temperature, salt stress

## Abstract

Low temperature and soil salinization during cotton sowing and seedling adversely affect cotton productivity. Exogenous melatonin (MT) can alleviate the damage caused to plants under non-biological stress; thus, applying MT is a means to improve the growth condition of crops under stress. However, achieving this goal requires a thorough understanding of the physiological regulatory mechanisms of MT on cotton seedlings under low temperature and salinity stress. This study could bring new knowledge on physio-biochemical mechanisms that improve the tolerance of cotton seedlings to combined effects of low temperature and salt stress using an exogenous foliar application of MT. The phytotron experiment comprised two temperature levels of cold stress and control and five MT treatments of 0, 50, 100, 150, and 200 μM and two salinity levels of 0 and 150 mM NaCl. Compared with the control treatments (non-salinity stress under cold stress and control), the combined stress of salt and low temperature reduced cotton seedlings’ biomass and net photosynthetic rate (*P_n_*), aggravated the membrane damage, reduced the potassium (K^+^) content, and increased the sodium (Na^+^) accumulation in the leaves and roots. Under NaCl stress, exogenously sprayed 50–150 μM MT increased the biomass and gas exchange parameters of cotton seedlings under salt and low temperature combined with salt stress, reduced the degree of membrane damage, and regulated the antioxidant enzyme, ion homeostasis, transport, and absorption of cotton seedlings. The pairwise correlation analysis of each parameter using MT shows that the parameters with higher correlation with MT at cold stress are mainly malondialdehyde (MDA), peroxidase (POD), and catalase (CAT). The highest correlation coefficient at 25 °C is observed between the K^+^ and Na^+^ content in cotton seedlings. The conclusion indicates that under salt and low-temperature stress conditions, exogenous application of MT primarily regulates the levels of *P_n_*, superoxide dismutase (SOD), andPOD in cotton seedlings, reduces Na^+^ and MDA content, alleviates damage to cotton seedlings. Moreover, the most significant effect was observed when an exogenous application of 50–150 μM of MT was administered under these conditions. The current study’s findings could serve as a scientific foundation for salinity and low-temperature stress alleviation during the seedling stage of cotton growth.

## 1. Introduction

Plants are often exposed to various environmental stresses, including drought, heavy-metal contamination, heat stress, low temperature, and soil salinization. Among these stresses, salinity and low temperature are important abiotic factors that are consequently associated with limited plant growth and productivity [1,2,3]. Moreover, over 50% of the world’s arable land is predicted to be affected by salinity by 2050 [4,5], and 15% of the suitable agricultural areas are affected by temperature stress worldwide [6]. Soil salinization is often accompanied by temperature stress because changes in ambient temperature are more frequent than changes in other abiotic factors. On the other hand, changes in ambient temperature rapidly aggravate other environmental stresses, including salinity [1,6].

Salinity is a worldwide concern that reduces agricultural output and imposes substantial revenue losses [7]. Salinity caused by NaCl has become a focus of environmental research investigations. NaCl-associated salinity leads to a wide range of changes in plant metabolism [8,9]. An increasing NaCl concentration affects plants in several ways. It causes nutrient deficiencies, osmotic stress, and specific ion toxicity, affecting several physiological mechanisms involved in plant metabolism [10]. Salt stress affects a range of important mechanisms, including photosynthesis, energy and lipid metabolism, and protein synthesis [11,12]. Low-temperature stress generally affects plant growth and induces reactive oxygen species production, damaging cell membrane structure [13]. Previous studies reported that low-temperature stress rapidly increases antioxidant activities, such as superoxide dismutase (SOD), peroxidase (POD), and catalase (CAT) [14,15].

Melatonin (N-acetyl-5-methoxytryptamine; MT), which was initially identified and isolated from the pineal glands of cows, is an important multifunctional hormone that is involved in modulating a wide range of animal physiological processes, such as sleep [16], immunity [17], circadian rhythm and antioxidant activity [18]. Melatonin was discovered for the first time in 1995 in vascular plants as an indoleamine hormone [19]. MT plays a key role in plant growth and development by exerting an auxin-like function [20,21]. The first plant phytomelatonin receptor (CAND2/PMTR1) was recently detected in *Arabidopsis thaliana*, allocating to melatonin the concept of phytohormone [22,23]. A new phytomelatonin receptor (named *ZmPMTR1*) that plays a key role in plant osmotic and drought stress tolerance was also identified in *Zea mays* [24]. Several studies revealed the importance of MT in mitigating biotic and abiotic stresses [25,26,27,28,29,30,31,32,33]. Known as an antioxidant and free radical scavenger, exogenously applied MT improved plants’ tolerance to biotic and abiotic stresses, which confers to plant stress resistance by enhancing photosynthesis, ion homeostasis, and antioxidant enzyme activities [34,35]. Exogenous foliar application of MT is involved in numerous physiological processes to improve plant resistance to salinity stress [36]. A recent study demonstrated that exogenously applied MT could sustain a high photosynthetic rate in tea plants, which enhances salt tolerance via its effects on antioxidant response against environmental stress [37]. Exogenous application of MT under salt stress showed alleviating effects on horticultural crops, such as an increase in primary root length and antioxidant activity in sunflower [38], an increase in net photosynthetic rate and stomatal conductance in tomato [39], and improving photosynthetic efficiency, endogenous melatonin content, and ion homeostasis in upland cotton [40]. On the other hand, exogenous application of melatonin was found to improve the resistance capacity of horticultural crops to low-temperature stress by increasing photosynthetic rate, antioxidant activity, and decreasing malondialdehyde (MDA) content in tea plants [41], increasing endogenous MT content, antioxidant activity, and decreasing MDA content in bermudagrass [42]; and increasing photosynthetic carbon assimilation, activities of antioxidant enzymes, levels of non-enzymatic antioxidants, anddecreasing the impact of cold on tomato membrane damage and ROS accumulation [39].

Cotton is the world’s most important natural textile fiber with an impressive economic value and is considered the backbone of the textile industry. Despite the capacity of cotton plants to tolerate environmental stress, exposure to stress conditions could negatively affect cotton growth, development, and yield [43,44]. When cotton is grown in arid/semiarid regions, seedlings are often exposed to both low-temperature and salinity stresses. Therefore, several investigations have aroused keen interest in improving cotton resilience. Previous studies revealed the mechanism of exogenous melatonin on crop resilience improvement under a single stress of either salinity or low temperature. Understanding the significance of exogenous MT on cotton seedlings’ physio-biochemical characteristics under the combined stress of salt and low temperature would have great theoretical and practical meaning. Therefore, the main purpose of this study was to determine the physiological and biochemical responses of low-temperature and NaCl-stressed cotton seedlings sprayed with exogenous melatonin.

## 2. Results

### 2.1. The Effect of MT on Cotton Seedling Biomass under Low Temperature and Salt Stress

Compared with the control, low temperature and salt stress, as well as their combined stress, significantly reduced the shoot dry weight, while root difference did not significantly affect it. Compared with the control (CK) at 15 °C, the coupled stress of salt and low temperature insignificantly affected the dry weights of the shoot and root of cotton seedlings (Figure 1). Under the salt stress condition at 25 °C, although the aboveground dry weight of cotton seedlings treated with exogenous MT increased slightly (compared with 0 μM MT), it was not significant, and there was no significant difference between different MT treatments. Under 25 °C, when compared to the 0 μM MT-saline treatment, exogenous MT significantly increased root dry weight, except for 200 μM MT; the highest value was obtained with 150 μM MT.

### 2.2. Effects of MT on Gas Exchange Parameters of Cotton Seedling Leaves under Low Temperature and Salt Stress

Leaf gas exchange parameters, including the net photosynthetic rate (*P_n_*), stomatal conductance (*g_s_*), intercellular carbon dioxide concentration (*C_i_*), and transpiration rate (*T_r_*), were measured for evaluating the physiological responses of cotton seedlings to MT under the combined stress. Salinity (0 μM at 25 °C) or low temperature (control at 15 °C) stress alone have significantly reduced *P_n_*, *g_s_*, *T_r_*, and *C_i_* of cotton seedlings, in comparison with the control treatment at 25 °C. Compared with control at 15 °C, the combined stress reduced the *P_n_*, *g_s_*, *T_r_*, and *C_i_* of cotton seedlings (Figure 2). Under the salt stress condition at 25 °C, with the increase in exogenous MT concentration, *P_n_*, *g_s_*, *T_r_*, and *C_i_* all showed a trend of increase firstly and then decreased in the 200 μM MT treatment compared with the non-MT treated saline treatment. Except for *C_i_*, the highest values of gas exchange parameters appeared in the 150 μM MT treatment. Under the combined stress of low temperature and salinity when compared to the MT-untreated saline, the application of different concentrations of exogenous MT insignificantly increased *P_n_*, *g_s_*, and *T_r_* of cotton seedlings, and with the increase in MT concentration, *P_n_*, *g_s_*, and *T_r_* also showed a trend of increasing and then decreasing. The highest value of *P_n_* was obtained with the 100 μM MT treatment.

### 2.3. The Effect of MT on Cotton Seedling Lipid Peroxidationunder Low Temperature and Salt Stress

Compared with control at 15 °C, low temperature plus salt stress did not significantly affect lipid peroxidation, measured as MDA content, nor the production rate of superoxide anion in leaves (Figure 3). All the concentrations of exogenous MT significantly decreased the MDA content under the combined stress of salt plus low temperature in comparison with the saline treatment in the absence of MT. Only the concentration of 100 μM MT significantly decreased the superoxide anion production rate under the combined stress of salt plus low temperature in comparison with the MT-untreated saline. At 25 °C, the exogenously applied MT (150 μM) significantly reduced both the superoxide anion production rate and MDA content in cotton seedling leaves under salt stress. At the same time (under 25 °C), compared with the saline treatment in the absence of MT, the superoxide anion production rate was insignificantly reduced by 29.6–4.4% under the 50–100 μM MT treatment, and the MDA content was significantly reduced by 43.3–67.7% under the 100–200 μM MT treatments, respectively. Under the combined stress of low temperature and saline conditions, spraying different concentrations of MT, compared with saline treatment in the absence of MT, the superoxide anion production rate significantly dropped to the value of 0.0034 μmol min^−1^ mg^−1^ port under the 150 μM MT treatment. With the increase in exogenous MT concentration, the content of MDA in leaves gradually decreased under the combined stress of low temperature and salt, and it significantly decreased to become stable when treated with 100–200 μM MT.

### 2.4. Effect of MT on Antioxidant Enzyme Activities under Low Temperature and Salt Stress

Table 1 shows the effect of exogenously applied MT on the antioxidant enzymes of cotton under salt stress alone or combined. Compared with controls at 15 °C, combined stress significantly reduced the contents of POD and CAT in leaves, and salt stress alone reduced SOD and POD activities. The combined stress of low temperature and salt did not affect the SOD activity in comparison with the control treatment. Compared with the saline treatment in the absence of MT at 25 °C, exogenously sprayed MT significantly reduced the SOD and POD activity and did not affect CAT activity in leaves. Compared with the saline treatment in the absence of MT under the combined stress of low temperature and salt, spraying 50–100 μM MT increased SOD and POD activity and decreased that of CAT; salt stress alone showed the same trend. Using an interactive analysis between the treatments, it was found that different temperature conditions had a significant effect on the SOD and CAT activities, but the POD of leaves under salt stress did not significantly affect them. Different MT concentrations significantly impacted POD and CAT while having an insignificant effect on the SOD of cotton seedlings under salt stress. The POD and CAT activities in cotton seedling leaves were affected by the interaction of temperature and MT.

### 2.5. Effect of MT on Ion Homeostasis and Absorption under Different Temperatures and Salt Stress

Results (Figure 4) show that compared with the control treatment at 25 °C, salt-stressed treatment alone significantly reduced the potassium (K^+^) ion content in the leaves and roots by 15% and 50% and increased the sodium (Na^+^) ion by 4.2 and 2.5 times-, respectively. The combined stress reduced the K^+^ content in the roots and significantly increased the accumulation of Na^+^ in the leaves and roots. After foliar spraying of different concentrations of MT, the K^+^ content in the roots was significantly increased when compared with the saline treatment in the absence of MT at 25 °C, and the K^+^ content in the leaves significantly increased with the exogenous foliar application of 150 μM when compared with the saline treatment in the absence of MT at 15 °C. The Na^+^ content in the leaves and roots was significantly reduced, but the difference between different concentrations of MT was insignificant. Under the combined stress conditions, compared with the saline treatment in the absence of MT, the application of exogenous MT to the K^+^ content in the leaves and roots of cotton seedlings did not significantly affect. Different concentrations of exogenous MT significantly reduced the content of Na^+^ in the leaves and roots of cotton seedlings when compared with the saline treatment in the absence of MT at 25 °C.

Compared with the control treatment, the coupled stress of low temperature and salt significantly increased the Na^+^/K^+^ ratio in the leaves and roots of cotton seedlings (5.4 and 2.9 times, respectively) (Figure 5A,B), and significantly increased the Na^+^ uptake roots (5.9 times) (Figure 5D). Exogenous foliar spraying of different MT concentrations significantly reduced the Na^+^/K^+^ ratio in the leaves and roots under the salt stress at 25 °C and correspondingly reduced the Na^+^ absorption. Under the combined stress, the exogenous foliar supplementation with MT, the Na^+^/K^+^ ratio in the leaves did not significantly affect. The Na^+^/K^+^ ratio in the roots significantly decreased with the foliar application of MT at 200 µM under the combined stress. On the other hand, compared with the control, the combined stress significantly increased the transfer of K^+^ ions from roots to shoots and then from shoots to leaves. After exogenously foliar spraying of different concentrations of MT, the low temperature plus salt stress significantly increased K^+^ transport from roots to aboveground parts. Compared with the saline treatment in the absence of MT, different concentrations of MT slightly reduced the transport of K^+^ from roots to shoots.

## 3. Discussion

Abiotic stresses such as salinity and low temperature severely limit the growth and development of plants. Moreover, nearly 50% of the annual yield loss of major crops worldwide is related to abiotic stresses [45]. The combined effect of low temperature and salt stress reduces leaf area, relative water content, water potential, transpiration rate, fresh weight, and dry weight of plant stems and roots [46]. To gain new knowledge of cotton seedling responses to exogenously applied MT under combined low temperature and salt stress, seedlings in a phytotron were foliar sprayed with MT to identify their physiological and biochemical responses. In this study, the combined effect of low temperature and salt stress significantly and negatively affected seedlings’ physiological and biochemical mechanisms (Table 1, Figure 1, Figure 2, Figure 3, Figure 4, Figure 5 and Figure 6). In agreement with our findings, previous studies have concluded that the effects of combined stress on crop growth and productivity may be devastating [47,48]. For example, drought and high temperature, salinity and high temperature, ozone and salinity, ozone and high temperature, nutrient stress and drought, nutrient stress and salinity, ultraviolet light and high temperature, ultraviolet light and drought, strong light and heat, drought or low temperature and other stress interactions showed significant negatives impacts on crop yield [48,49]. Literature reported It has been reported that combined stress such as drought plus high-temperature stress has a more serious negative impact on the number of wheat tillers, chlorophyll content, yield, and harvest index than a single stress of drought or high temperature [50,51]. In this study, the combination of low temperature plus salinity stress has a greater impact on cotton seedling biomass when compared to the control treatment (Figure 1). This is mainly related to the senescence and dehydration of plant cells under abiotic stress, and salinity limits cell elongation and division, which leads to a decrease in the growth rate of roots and shoots and a decrease in dry matter accumulation [52]. After exogenous foliar supplementation with MT under the different stresses of low temperature and salinity plus low temperature, the aboveground and underground biomass remained unaffected, while the underground biomass was significantly increased by the foliar supplementation with MT (50, 100 and 150 µM) under salt stress at 25 °C in comparison with the saline treatment in the absence of MT (Figure 1). This is related to the ability of MT to regulate the physical process of cell wall extension to induce plant root growth and promote the accumulation of dry matter [53].

Photosynthesis is the basis of plant growth and development, but it is susceptible to environmental stresses (particularly sensitive to low-temperature stress). The main components of photosynthesis (electron transport, Calvin cycle, stomatal conductance) are negatively affected under low-temperature stress [46]. In a previous study, we observed that sustained leaf gas exchange parameters, including *P_n_*, g_s_, *T_r_*, and *C_i_* are essential for cotton seedlings’ survival under salt stress conditions [43]. Regulation of leaf gas exchange parameters is reported to be an important aspect of improving crop resilience to different biotic and abiotic stress conditions [54]. In this study, *g_s_* and *T_r_* decreased in response to salinity alone, low temperature alone, and salinity plus low temperature, and plants reduced water loss by closing their stomata. The closing of stomata leads to insufficient CO_2_ and inhibits photosynthesis (Figure 2). This is consistent with the research results of Chatrath et al. [55] who reported that plants may close their stomata under saline conditions to reduce water loss. As shown in Figure 7B, under the combined stress of salinity plus low temperature the antioxidant enzyme activity of cotton seedlings unilaterally affects photosynthetic parameters. Biomass is not directly affected by photosynthetic parameters; membrane damage and other parameters have no mutual influence. After the foliar application of MT (100 and 150 µM) under salt stress at 25 °C, the *g_s_* and *T_r_* of cotton seedlings were increased, thereby increasing their *P_n_*, but the foliar application of MT did not alter photosynthesis parameters under the combined effect of salinity plus low temperature (Figure 2). This is in contradiction with the research results of Irshad et al. [56] and Zhang et al. [33], who observed that exogenous MT significantly improved *P_n_* under low-temperature stress and salt stress, respectively. This difference between our findings and those of previous studies is probably due to (1) the different species used as plant material (*Medicago truncatula* in a previous study and *Gossypium hirsutum* L. in the current study), (2) the MT application mode (previous studies applied MT by seed priming and or hand MT application one week before cold stress induction, while foliar MT treatment and low-temperature stress were conducted simultaneously in this study).

The translocation of substances and ions in plants mainly depends on the integrity of the structure and function of the cellular membranes. When the plant is subjected to abiotic stress, the plant’s cellular membrane system is the first to be impacted and damaged. One of the damages caused by abiotic stress on cell membranes is lipid peroxidation. The final product of peroxidation damage to cell membranes by reactive oxygen species is MDA. The production of reactive oxygen species (ROS) such as H_2_O_2_, superoxide anion, hydroxyl anion, and MDA under salt conditions severely destroys chlorophyll content, destroys lipids and mitochondria, and causes plant necrosis [33]. In this study, under the combined stress, cotton seedlings’ MDA content increased compared with the control, although the superoxide anion production rate remained unchanged compared with the control, which led to membrane lipid peroxidation (Figure 3). The research results are consistent with those of Parveen et al. [57]. Abiotic stress causes an increase in the level of active oxygen in the plant body and simultaneously activates the defense system to reduce the harm of active oxygen. SOD, CAT, and POD are the key enzymes to remove active oxygen in cells, and the ability to remove active oxygen depends on the coordination of these enzymes [58]. As the first line of defense for ROS removal, SOD can disproportionate superoxide anions into O_2_ and H_2_O_2_, and POD and CAT can remove H_2_O_2_ in peroxisomes [59]. In this experiment, compared with the control at 15 °C, salt stress alone as well as the coupled stress of salt plus low temperature decreased the CAT and POD activities. Furthermore, the low-temperature stress increased the POD and CAT activities and decreased the SOD activity when compared to the control treatment at 25 °C (Table 1). MT can increase the activity of antioxidant enzymes to alleviate the damage caused by abiotic stress on plants. Studies have shown that the application of exogenous MT can increase the SOD, POD, and CAT activities in cucumber under salt-stress environments [60] and increase the SOD activity in wheat [61] and cucumber [62] under low-temperature stress. In addition, under salt and heat stress, applying MT increased tomato CAT activity and reduced SOD activity [63]. In this experiment under the combined stress of salinity plus low temperature, after applying the exogenous MT, compared with the salt treatment without the exogenous MT, the exogenous spraying of MT at a concentration of 200 μM decreased the POD activity of cotton seedlings. In contrast, under the combined stress of salinity plus low temperature, spraying of different concentrations of MT significantly increased the SOD activity of cotton seedlings (Table 1). Using interactive analysis, it is known that temperature has a very significant impact on SOD and CAT, and MT has a very significant impact on POD and CAT. At the same time, POD and CAT are affected by the interaction of temperature and MT.

Under salt stress, the excessive accumulation of Na^+^ in plant cells leads to the production of ROS, which breaks the dynamic balance of the scavenging system and initiates membrane lipid peroxidation and degreasing. The membrane structure is destroyed, and the MDA content increases [58]. In this study, after MT under salt stress, the opposite conclusion was reached and found that the Na^+^ content in cotton seedlings was reduced, the activity of antioxidant enzymes, the degree of lipid peroxidation was increased, the degree of lipid peroxidation damage was reduced, and the photosynthetic parameters and dry matter content were increased. Previous studies reported that the high accumulation of Na^+^ in the apoplasts of roots and stems easily triggers osmotic stress [64]. The higher accumulation of Na^+^ leads to an increase in the ratio of Na^+^/K^+^ ratio and promotes ion imbalance, thereby hindering metabolic functions and destroying plant activities [33]. For example, photosynthesis declines, and dry matter accumulation decreases, which ultimately leads to reduced crop growth [53], production of reactive oxygen species [65], and plant damage. The results of this study showed that, compared with the control at 25 °C, salinity reduced the K^+^ accumulation in the leaves and roots of cotton seedlings and significantly increased the accumulation of Na^+^, resulting in a significant increase in the ratio of Na^+^/K^+^ and at the same time increased the absorption of Na^+^ (Figure 4 and Figure 5). However, after spraying exogenous MT at 25 °C under saline conditions, Na^+^ accumulation in leaves and roots was significantly reduced. The K^+^ accumulation in the leaves was increased, and the Na^+^/K^+^ ratio was reduced. Since the K^+^ dynamic balance is affected by the accumulation of Na^+^, an appropriate Na^+^/K^+^ should be maintained to improve the salt tolerance of crops [66]. Studies have found that under abiotic stress, plants, by closing stomata and reducing transpiration, may limit the transport of Na^+^ to plant tissues [67]. The results of this study indicate that compared to salt stress, the stomatal conductance is reduced under the combined stress of low temperature and salt, and the Na^+^/K^+^ ratio is lower than that of salt stress. In the current study, using interactive analysis, temperature was shown to significantly affect K^+^, Na^+^, Na^+^/K^+^ ratio, Na^+^ absorption, and K^+^ transport in roots and leaves, whereas Na^+^ content in leaves was not affected. The influence of exogenous MT on K^+^ content is insignificant. The Na^+^ content in root removal is not affected by the interaction of temperature and MT, and the others are all affected by the interaction of temperature and MT.

Studies have shown that low-temperature stress can enhance the antioxidant defense ability by regulating the metabolic and photosynthetic reactions of tea plants, wheat, Arabidopsis, and other plants, thereby improving the cold tolerance of plants [37,68]. The results of this study showed that when exogenous MT is applied under low temperature plus salt stress, the ion content directly affects the protective enzymes, photosynthetic parameters, membrane damage, and biomass, while the protective enzymes and photosynthetic parameters negatively affect the ion content. Protective enzymes affect membrane damage in one direction and have a two-way effect on photosynthesis (Figure 7). In summary, using interactive analysis, we have shown that temperature has a significant effect on the shoots and root biomasses under salt stress conditions. Exogenous MT successfully increased the biomass and physiological and biochemical indexes of cotton seedlings under the combined stresses. In addition, reducing the absorption of ROS and Na^+^ in plants is the focus of this study, indicating that exogenous MT has the potential to restore crop growth under salt stress. The relationships between various parameters of cotton seedlings at 15 °C and 25 °C under salt stress conditions are shown in Figure 6 and Table 2. It was observed that improving antioxidant enzyme activities with exogenous MT results in an improvement in leaf gas exchange parameters and K^+^ uptake while decreasing Na^+^ uptake.

Expression of the phythomelatonin receptor (*ZmPMTR1*) induced using abiotic stresses, including low temperature and salt, suggests its potential function in maize responses to various abiotic stresses and may be useful for crop genetic improvement [24]. Recently, various studies aimed to investigate the function of melatonin as a signaling molecule that is involved in plant growth regulation and stress responses, mainly the identification and analysis of its receptors in various plant species [28,69]. More in-depth studies on cotton are required in the future perspectives on the MT-receptor interaction (identification, location, regulation, etc.). Exogenous MT has different regulatory mechanisms on the ion content, photosynthetic parameters, membrane damage, protective enzyme activities, and biomass accumulation of cotton seedlings under coupled salt and low-temperature stress (Figure 7). Protecting enzymes under salt stress affects photosynthesis in two directions while affecting photosynthesis in one direction under low temperatures plus salt stress. The antioxidant enzyme activity under salt stress affects membrane damage one way, and ion content one-way affects membrane damage and biomass, while under the combined stress of low temperature plus salt, the antioxidant enzymeand membrane damage, ion content, and membrane damage and biomass have two-way effects. Under double coercion, MT’s adjustment of the above parameters is more complex.

## 4. Materials and Methods

### 4.1. Plant Material and Experimental Design

The cotton variety, Xinluzhong37, was purchased from The Seed Industry Co., Ltd., Alar City, China. It is a mid-early maturing upland cotton variety with a growth period of about 140 days. Melatonin (MT), sodium chloride (NaCl), sodium sulfate (Na_2_SO_4_), and other chemical reagents are all analytically pure, purchased from Beijing Soleibao and Sinopharm Chemical Reagent Co., Ltd. (Shanghai, China).

The experiment was carried out in a phytotron in the Xinxiang Comprehensive Experimental Station of the Chinese Academy of Agricultural Sciences (35.09° N, 113.48° E, and altitude 81 m). The humidity of the climate room is 40–50%, the photoperiod is 12 h, the light intensity is 600 μmol m^−2^ s^−1^ (lighting is provided using LED lights), and the temperature is controlled according to the experimental design. The experimental design is shown in Table 3.

Uniform size and plump cotton seeds were selected, disinfected with 1000 times diluted solution of carbendazim, and rinsed with deionized water after 30 min. The sterilized seeds were sown at a depth of 2 cm in PVC pots (diameter 6 cm, height 24 cm) containing 780 g of fine sand (the sand was sterilized at high temperature). After sowing, the PVC pots were covered with an opaque shading board to keep the surface of the sand moist and facilitate the germination of seeds. For germination, the PVC pots were placed in a phytotron with a day/night temperature of 25/20 °C. After the seeds germinated, the shading was removed from the pots. Seven (7) plants per treatment with one plant/pot were used in this experiment, with a total number of 70 plants, and three (3) replications per treatment were used for all the measurements. When the seedlings grew to one true leaf, seedlings were watered with 1/2 Hoagland nutrient solution [9]. The pH of the nutrient solution was between 5.8 and 6.2; each pot was watered every five days with about 80 mL of the 1/2 Hoagland solution. When the seedlings were grown to two leaves and one heart leaf, they were randomly divided into groups to start the salt and temperature treatments. For room temperature (25 °C) and low temperature (15 °C) treatments, seedlings were placed in phytotrons with day/night temperatures of 25/18 °C and 15/8 °C, respectively. On the 1st, 4th, and 9th day under the temperature treatment, seedlings were irrigated with 150 mmol L^−1^ salt water (NaCl: Na_2_SO_4_ = 9:1) in the morning. Every day after turning off the lights, different concentrations of MT (0, 50, 100, 150, 200 μmol L^−1^) were sprayed on the upper surface of the leaves of salt-stressed cotton seedlings at 15 and 25 °C. On the 10th day, the fully expanded leaves were sampled for the measurement of various indicators.

### 4.2. Measurements

#### 4.2.1. Determination of Cotton Seeding Biomass and Gas Exchange Parameters

Three plants were randomly selected for each treatment, the shoots were separated from the roots, and the fresh weights were, respectively weighed and put into sample bags. The samples were put into a sample bag at 105 °C for 30 min and oven-dried at 75 °C for 48 h; the dry weight was then measured.

The LI-6400XT photosynthesis measuring system (Licor, Lincoln, NE, USA) was used to measure the net photosynthetic rate (*P_n_*), stomatal conductance (*g_s_*) intercellular carbon dioxide concentration (*C_i_*), and transpiration rate (*T_r_*) of the upper fully expanded leaves of cotton seedlings between 9:00–11:00 in the morning with photosynthetically active radiation (PAR) at 1000 μmol·m^−2^·s^−1^.

#### 4.2.2. Determination of MDA, Superoxide Anion and Antioxidant Enzyme Activities

About 0.3 g of fresh leaf tissue was ground using a pestle and mortar. An amount of 3.0 mL of 0.05 M precooled phosphate buffer (pH 7.8) was added to the homogenate, which was then centrifuged at 15,000× *g* at 4 °C for 20 min. Lipid peroxidation was measured spectrophotometrically from MDA content using a thiobarbituric acid (TBA) reaction following the method described by Heath and Packer [70]. At the same time, the same supernatant was used to determine the leaves’ superoxide anion production rate [71].

About 0.3 g of fresh leaf tissue was ground using a pestle and mortar. An amount of 2.7 mL of 0.05 M precooled phosphate buffer (pH 7.8) was added to the homogenate, which was then centrifuged at 15,000× *g* at 4 °C for 20 min. After obtaining the supernatant, refer to the following method to determine the protective enzyme activity. SOD activity was determined spectrophotometrically from the inhibition of the photochemical reduction of nitroblue tetrazolium (NBT) at 560 nm [72]. POD activity was measured using the determination of guaiacol oxidation by H_2_O_2_ at 470 nm [73]. CAT activity was measured by monitoring the disappearance of H_2_O_2_ at 240 nm [74].

#### 4.2.3. Determination of Seeding Biomass and Ion Content

On the 10th day after the treatment, three cotton seedling leaves and roots were randomly selected for each treatment, oven-dried at 105 °C for 30 min, and then continued drying at 75 °C for 48 h; the dry weight was then measured. The dried leaves and roots were ground into powder. Each sample, weighing 0.15 g, was digested with H_2_SO_4_-H_2_O_2_ and diluted in a 100-mL volumetric flask. The filtered supernatant was used to determine the Na^+^ and K^+^ ion contents in the leaves and roots of cotton seedlings using a flame photometer. Na^+^ uptake at the seedlings’ root surface and K^+^ translocation from root to shoot were computed by following the equations described by Shabala and Mackay [75,76], respectively.
(1)$$\mathrm{Na}\;\mathrm{ion}\;\mathrm{uptake}\;\mathrm{at}\;\mathrm{root}=\frac{Sum\;of\;Na\;concentration\;in\;tissues}{Total\;root\;dry\;wright\;of}$$
(2)$$Translocation\;factor=\frac{K\;concentration\;in\;leaves}{K\;concentrantion\;in\;roots}$$

### 4.3. Statistical Analysis

The data were statistically analyzed using one-way ANOVA using EXCEL 2020 and DPS V13.5 software. All experimental data were expressed as means ± standard deviation. All treatment means (*n* = 3) were compared for any significant differences using Duncan’s multiple range tests at *p* < 0.05. Data fitting and graphical presentation were carried out in Origin-Pro 2021a (Origin Lab, Northampton, MA, USA). A general linear regression model was used to fit the relationships between parameters.

## 5. Conclusions

The current study demonstrated that exogenous foliar application of MT significantly affects the physiological and biochemical characteristics of cotton subjected to combined stress of salinity and low temperature. Under salt stress, the spraying of 50 to 150 μM MT on the leaf surface increased the biomass, *P_n_*, and SOD and POD activities while reducing membrane damage of cotton seedling leaves. The exogenously supplied MT reduced the accumulation of Na^+^ and MDA under the coupled stress of low temperature and salt. The regulatory mechanism of exogenously applied MT remains different under salt stress alone as compared to the combined salinity and low-temperature stress. The regulation of the mechanism under the combined stress of low temperature and salinity is more complex. Further study is necessary to investigate the molecular mechanisms of MT-regulated salinity and low-temperature stress tolerance in cotton.

## Figures and Tables

**Figure 1 plants-12-03730-f001:**
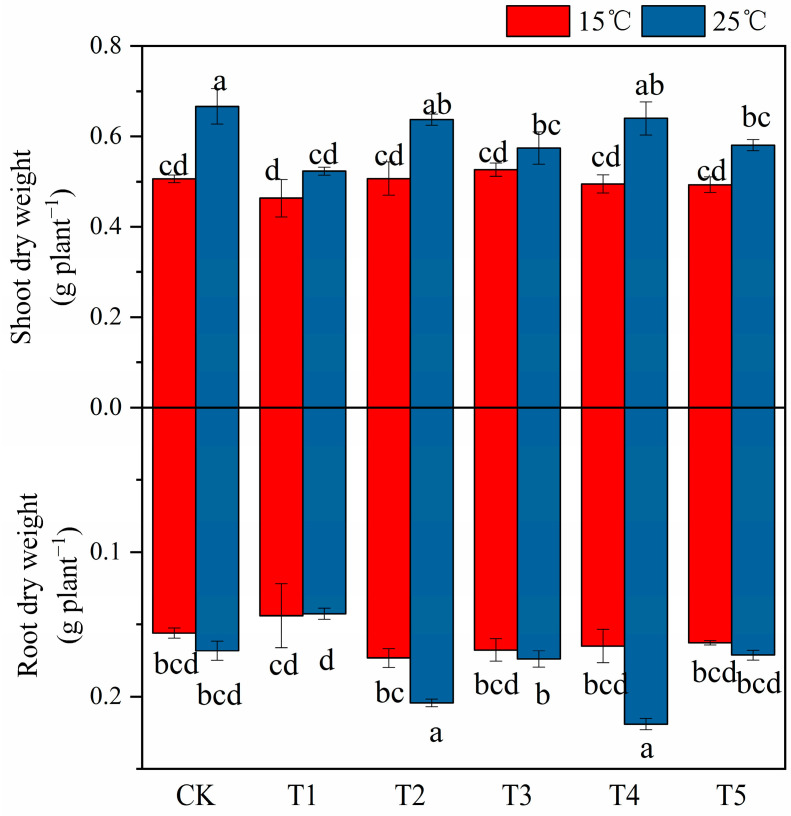
Effects of exogenous melatonin (MT) on shoot dry weight and root dry weight of 150 mM NaCl−stressed cotton seedlings under temperatures of 15 °C and 25 °C. Note: CK = Control (0 μM salt0 μM MT), T1 = 150 mM NaCl+0 μM (MT), T2 = 150 mM NaCl+50 μM (MT), T3 = 150 mM NaCl+100 μM (MT), T4 = 150 mM NaCl+150 μM (MT), T5 = 150 mM NaCl+200 μM (MT). Values are means ± standard deviation (*n* = 3). Different letters represent significant differences at *p* < 0.05 level between the experimental treatments.

**Figure 2 plants-12-03730-f002:**
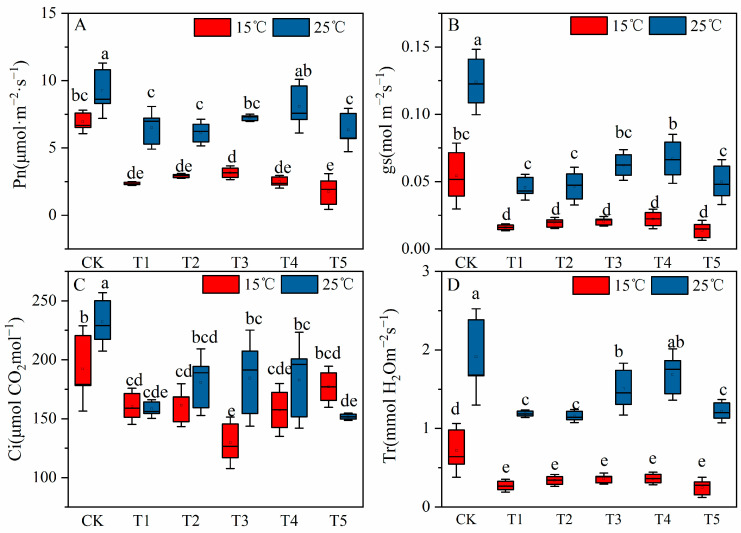
Effects of exogenous melatonin (MT) on (**A**) net photosynthetic (*P_n_*), (**B**) stomatal conductance(*g_s_*), (**C**) transpiration rate (*T_r_*), and (**D**) intracellular CO_2_ concentration (*C_i_*) of cotton seedlings under the condition of 150 mM NaCl and temperatures of 15 °C and 25 °C. Note: CK = Control (0 μM salt+0 μM MT), T1 = 150 mM NaCl+0 μM (MT), T2 = 150 mM NaCl+50 μM (MT), T3 = 150 mM NaCl+100 μM (MT), T4 = 150 mM NaCl+150 μM (MT),T5 = 150 mM NaCl+200 μM (MT). Values are means ± standard deviation (*n* = 3). Different letters represent significant differences at *p* < 0.05.

**Figure 3 plants-12-03730-f003:**
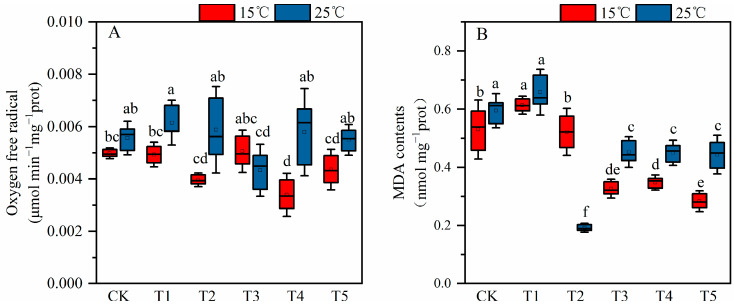
Effects of exogenous melatonin (MT) on (**A**) Superoxide anion content and (**B**) malondialdehyde (MDA) content of cotton seedlings under 150 mM NaCl-stressed and temperatures of 15 °C and 25 °C. Note: CK = Control (0 μM salt+0 μM MT), T1= 150 mM NaCl+0 μM (MT), T2 = 150 mM NaCl+50 μM (MT), T3 = 150 mM NaCl+100 μM (MT), T4 = 150 mM NaCl+150 μM (MT),T5 = 150 mM NaCl+200 μM (MT). Values are means ± standard deviation (*n* = 3). Different letters represent significant differences at *p* < 0.05.

**Figure 4 plants-12-03730-f004:**
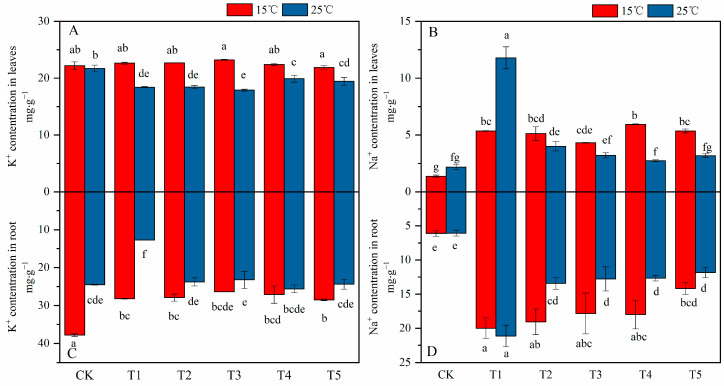
Effects of exogenous melatonin (MT) on (**A**) K^+^ concentration in the leaves, (**B**) Na^+^ concentration in the leaves, (**C**) K^+^ concentration in the roots, and (**D**) Na^+^ concentration in the roots of cotton seedlings under the 150 mM NaCl-stressed and temperatures of 15 °C and 25 °C. Note: CK = Control (0 μM salt+0 μM MT), T1 = 150 mM NaCl+0 μM (MT), T2 = 150 mM NaCl+50 μM (MT), T3 = 150 mM NaCl+100 μM (MT), T4 = 150 mM NaCl+150 μM (MT),T5 = 150 mM NaCl+200 μM (MT). Values are means ± standard deviation (*n* = 3). Different letters represent significant differences at *p* < 0.05.

**Figure 5 plants-12-03730-f005:**
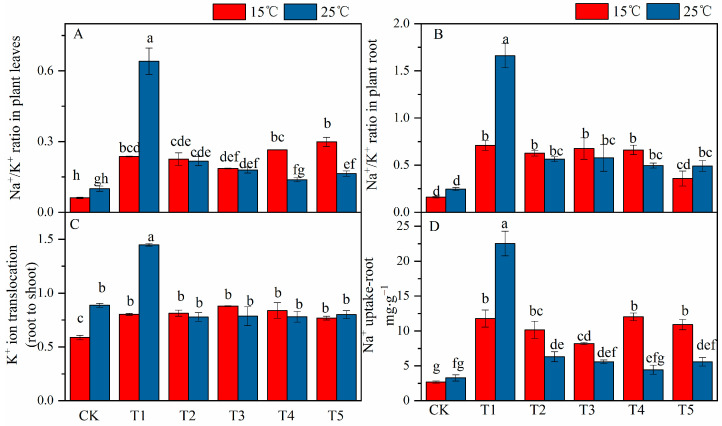
Effects of exogenous Melatonin (MT) on (**A**) Na^+^/K^+^ ratio in plant leaves, (**B**) Na^+^/K^+^ ratio in plant roots, (**C**) K^+^ ion translocation (root to shoot), and (**D**) Na^+^ uptake in-root of cotton seedlings under the 150 mM NaCl-stressed and temperatures of 15 °C and 25 °C. Note: CK = Control (0 μM salt+0 μM MT), 0 μM = 150 mM NaCl+0 μM (MT), 50 μM = 150 mM NaCl+50 μM (MT), 100 μM = 150 mM NaCl+100 μM (MT), 150 μM = 150 mM NaCl+150 μM (MT), 200 μM = 150 mM NaCl+200 μM (MT).Values are means ± standard deviation (*n* = 3). Different letters represent significant differences at *p* < 0.05.

**Figure 6 plants-12-03730-f006:**
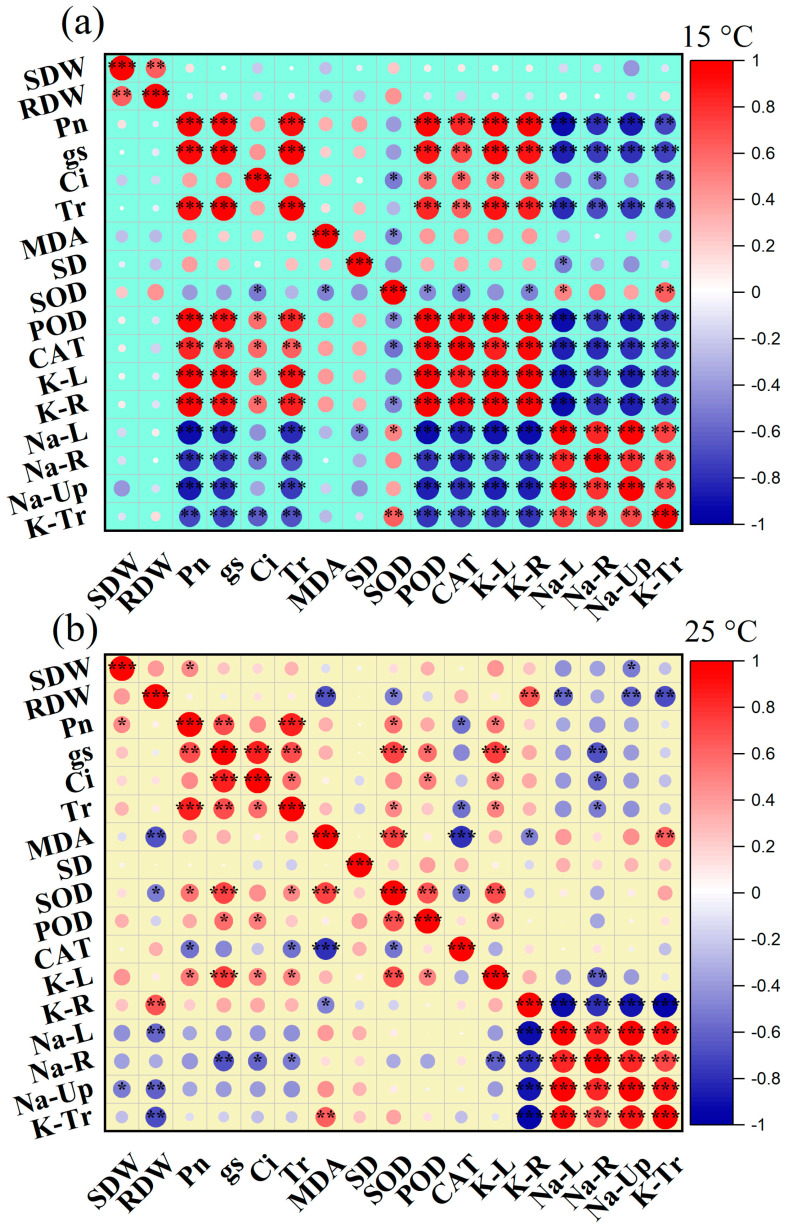
Correlation analysis of exogenous MT on various parameters of salt-stressed cotton seedlings under (**a**) low temperature (15 °C) and (**b**) 25 °C. SDW, shoot dry weight; RDW, root dry weight; *Pn*, net photosynthetic rate; *g_s_*, stomatal conductance; *C_i_*, intracellular CO_2_ concentration; *T_r_*, transpiration rate; K-L, leaf K^+^ content; K-R, root K^+^ content; Na-L, leaf Na^+^ content; Na-R, root Na^+^ content; Na^+^ uptake; K^+^ translocation. * *p* < 0.05, ** *p* < 0.01 and *** *p* < 0.001.

**Figure 7 plants-12-03730-f007:**
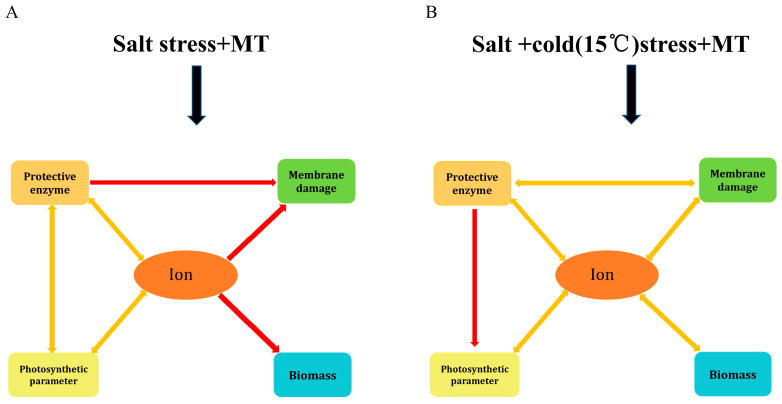
Effects of exogenous MT on the physiological mechanism of cotton seedlings under (**A**) salt stress and (**B**) under coupled low temperature and salt stress. The yellow double-orientation arrow indicates that the two affect each other; the red one-way arrow indicates unilateral influence.

**Table 1 plants-12-03730-t001:** Response of antioxidant enzymes activities to MT in cotton seedlings under temperatures of 15 °C and 25 °C and salt stress. Values are means ± standard deviation.

	Antioxidant Enzyme Activities in Leaf Tissues					
Treatment	SOD/U × g^−1^ × FW		POD/U × g^−1^ × FW		CAT/U × g^−1^ × FW	
	15 °C	25 °C	15 °C	25 °C	15 °C	25 °C
CK	67.32 ^cd^	108.32 ^b^	13.32 ^a^	7.38 ^b^	0.46 ^a^	0.01 ^b^
T1	69.27 ^cd^	78.52 ^c^	4.38 d ^e^	5.09 ^cd^	0.11 ^b^	0.02 ^b^
T2	147.18 ^a^	45.32 ^f^	5.17 ^cd^	5.95 ^c^	0.03 ^b^	0.07 ^b^
T3	137.57 ^a^	51.96 ^ef^	4.46 ^de^	3.55 ^ef^	0.04 ^b^	0.02 ^b^
T4	140.41 ^a^	58.09 d ^ef^	4.45 ^de^	3.79 ^ef^	0.04 ^b^	0.02 ^b^
T5	101.91 ^b^	63.78 ^de^	3.14 ^f^	3.95 ^ef^	0.03 ^b^	0.04 ^b^
Temp	11.79 **		1.43		29.48 **	
MT	2.60		1.46 **		20.29 **	
Temp× MT	1.35		9.90 **		33.43 **	

Note: CK = Control (0 μM salt + 0 μM MT), T1 = 150 mM NaCl + 0 μM (MT), T2 = 150 mM NaCl + 50 μM (MT), T3 = 150 mM NaCl + 100 μM (MT), T4 = 150 mM NaCl + 150 μM (MT), T5 = 150 mM NaCl + 200 μM (MT). Treatments: 15 °C, 25 °C; CK, control. Temp is temperature; MT is melatonin. Values are means ± standard deviation (*n* = 3). Different letters represent significant differences at *p* < 0.05. ** *p* < 0.01.

**Table 2 plants-12-03730-t002:** Stepwise regression of parameters of salt-stressed cotton seedlings at 15 °C and 25 °C under exogenous melatonin.

Independent Variable		Low Temperature (15 °C) + Salt Stress (150 mM)			25 °C + Salt Stress (150 mM)		
		Stepwise Regression Equation	R^2^	*p*	Stepwise Regression Equation	R^2^	*p*
X1	shoot dry weight	y = 1.443x2 + 0.266	0.40	0.01	y = 0.20x12 + 0.192x14 − 0.101x16 + 0.147	0.87	0.00
X2	root dry weight	y = 0.278x1 + 0.023	0.40	0.01	y = −0.070x17 + 0.244	0.47	0.00
X3	*P_n_*	y = −0.014x5 + 3.090x6 + 0.398x10 + 2.046	0.97	0.00	y = 3.558x6 + 2.126	0.72	0.00
X4	*g_s_*	y = −0.002x12 + 0.001x10 + 0.061x6 + 0.034	1.00	0.00	y = 0.002x13 + 0.001x9 + 0.000x5−0.108	0.92	0.00
X5	*C_i_*	y = −135.623x17 − 269.164	0.39	0.06	y = 891.435x4 + 122.900	0.72	0.00
X6	*T_r_*	y = 0.025x12 − 0.016x10 + 15.698x4−0.468	0.99	0.00	y = 0.203x3−0.033	0.72	0.00
X7	MDA	y = −0.002x9 + 0.635	0.23	0.04	y = −0.070x10 + 0.007x9 + 0.209x17 + 0.120	0.92	0.00
X8	Superoxide anion	y = 0.000x14 − 0.005	0.26	0.03	/	/	/
X9	SOD	y = 205.005x17 − 49.711	0.40	0.01	y = −1.211x15 + 7.685x10 + 101.543x7 − 1.916	0.96	0.00
X10	POD	y = 0.125x15 + 130.185x4 + 2.404x11 − 0.422	0.98	0.00	y = −4.530x7 + 32.189x11 + 0.083x9 + 0.531	0.85	0.00
X11	CAT	y = −0.052x15 − 2.979x6 + 0.329x10 − 0.498	0.96	0.00	y = 0.002x14 − 0.114x7 + 0.073	0.73	0.00
X12	K^+^-L	y = −0.057x16 + 0.042x15 − 35.334x4 + 0.604x13 + 22.223x17 − 11.876	0.96	0.00	y = 37.733x4 + 16.815	0.57	0.00
X13	K^+^-R	y = 0.42x11 + 48.308x4 + 1.020x12 − 33.413x17 + 31.137	1.00	0.00	y = 0.237x16 + 1.078x12−23.081x17 + 20.755	0.99	0.00
X14	Na^+^-L	y = 4.983x2 + 9.996x1 + 0.492x16 − 5.787	0.99	0.00	y = −0.896x7 + 4.021x1 + 0.534x16 − 1.744	1.00	0.00
X15	Na^+^-R	y = 0.758x13−4.227x11 + 4.298x12 + 2.025x14 − 110.436	0.87	0.00	y = 17.741x17 − 0.150x9 + 6.932	0.91	0.00
X16	Na^+^-uptake	y = −0.993x2 − 20.381x1 + 2.027x14 + 11.795	1.00	0.00	y = 1.713x7 − 7.595x1 + 1.869x14 + 3.308	1.00	0.00
X17	K^+^ translocation	y = 0.011x11 + 1.427x4 + 0.031x12 − 0.030x13 + 0.912	1.00	0.00	y = 0.119x7 + 0.012x16 + 0.040x12 − 0.039x13 + 0.856	0.99	0.00

Note: SDW, shoot dry weight; RDW, root dry weight; *P_n_*, net photosynthetic rate; *g_s_*, stomatal conductance; *C_i_*, intracellular CO_2_ concentration; *T_r_*, transpiration rate; K-L, leaf K^+^ content; K-R, root K^+^ content; Na-L, leaf Na^+^ content; Na-R, root Na^+^ content; Na^+^ uptake; K^+^ translocation.

**Table 3 plants-12-03730-t003:** The detailed experimental treatments.

Temperature	Treatment Label	NaCl Dose (mM)	MT Dose (μM)
15 °C	CK	0	0
	T1	150	0
	T2	150	50
	T3	150	100
	T4 T5	150 150	150 200
25 °C	CK	0	0
	T1	150	0
	T2	150	50
	T3	150	100
	T4 T5	150 150	150 200

## Data Availability

Not applicable.

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
