# Peer review of "Foliar Application of Melatonin Positively Affects the Physio-Biochemical Characteristics of Cotton (Gossypium hirsutum L.) under the Combined Effects of Low Temperature and Salinity Stress"

_plants, 2023, doi:10.3390/plants12213730_

Round 1
Reviewer 1 Report
Comments and Suggestions for Authors
Reviewer’s Recommendation: Major Revision
Reviewer’s comments to Authors
Authors should revise the manuscript carefully in light of below comments...............
1. Grammatical errors are present, please revise the whole manuscript to remove any possible grammatical and typos errors.
2. Error in sentence formation, please revise the whole manuscript to avoid the use of long sentences and confusing sentences/paragraphs.
3. Please maintain uniformity while in-text citation and referencing in the entire manuscript.
4. The reference does not meet the format requirements of the Journal so please check the references as per the authors guideline of the Journal.
5. It is advised to check and avoid too many self-cited papers.
6. The beginning of a new paragraph should be after some space, check in complete manuscript.
7. Throughout the whole manuscript the plant names should be in italic format.
8. This paper lacks final revision by the author as many general repetitions, typos, grammatical, sentence formation errors were found in the manuscript. It is not possible to mention all such errors. Thus revise the manuscript accordingly.
Abstract:
Authors should revise the manuscript carefully in light of below comments...............
- An abstract must be fully self-contained and make sense by itself, without further reference to outside sources or to the actual paper. It is important to provide the relevance or importance of your work and the main outcomes. Please revise the abstract accordingly.
- The abstract is not clear and the objective of the paper is not clearly validated from the abstract.
- The future perspective of the experiment should be mentioned in the abstract.
- The abstract should appropriately over the contents of the manuscript.
- In the keywords, it is strongly advisable to use suitable words that can aid in finding out the manuscript in current registers or indexes. Strictly avoid the use of title words in the keywords.
6. A graphical abstract is recommended for better perception of the present study.
7. A novelty statement is also encouraged to be added in the manuscript for bringing out the uniqueness of your study and its importance.
8. Please ensure that “Highlights” of the present work should be clearly mentioned in the manuscript.
Introduction:
Authors should revise the manuscript carefully in light of below comments...............
1. The literature from past work done in the same field missing to strengthen the introduction section. The need and importance of the present work should be clearly written in the introduction section.
2. The new aspects and innovations of this manuscript should be clearly and briefly described in this section.
3. The present state of knowledge in the subject should be described in introduction.
4. The literature should be sufficiently critical, current, and internationally evaluated.
Materials and Methods:
Authors should revise the manuscript carefully in light of below comments...............
1. Please try to merge the different sub-sections of the methodology as an individual mention for each component seems a bit unscientific method.
2. The size of manuscript seems to be large. It should be crisp and appropriate. Please revise it.
3. The text presented across the manuscript should be simple so that the scientist/workers in other disciplines will understand. Please revise it.
4. The different sections of manuscript are poorly cited with the references and required to update and validation with previous studies. The relevant papers listed below may be considered to enhance the scientific quality of manuscript significantly.
· Mariyam, S. et al. (2023). Review on nitric oxide at the forefront of rapid systemic signaling in mitigation of salinity stress in plants: crosstalk with calcium and hydrogen peroxide. Plant Science, 111835. https://doi.org/10.1016/j.plantsci.2023.111835
· Agnihotri, A. et al. (2018). Counteractive mechanism (s) of salicylic acid in response to lead toxicity in Brassica juncea (L.) Czern. cv. Varuna. Planta, 248, pp.49-68. https://doi.org/10.1007/s00425-018-2867-0
· Prajapati, P. et al. (2023). Nitric oxide mediated regulation of ascorbate-glutathione pathway alleviates mitotic aberrations and DNA damage in Allium cepa L. under salinity stress. International Journal of Phytoremediation, 25(4), pp.403-414. https://doi.org/10.1080/15226514.2022.2086215
Results and Discussion:
Authors should revise the manuscript carefully in light of below comments...............
- The results and discussion section needs to be elaborated more. The results should be clearly described in light of available knowledge and hypothesis and must be strongly validated with previous reports in the related subject area.
- The non-significant results was not clearly validated from the previous papers.
- Please carefully check, verify, and correct the results of the present experiments from the tables/figures/graphs provided in the manuscript.
- The discussion does not describe the results with proper facts and even does not validate the result with appropriate references. Please enrich it significantly.
- The discussion did not provide a specific reasons for the results. The provided explanation should be strengthen significantly.
- The strong hypothesis, scientific facts, and validation of previous reports are entirely missing. Please revise it.
Conclusion:
Authors should revise the manuscript carefully in light of below comments...............
- The conclusion section failed to enlighten the spirit of the present finding or work so required to revise it accordingly.
- In the conclusion section the authors have only mentioned the data but major finding is missing from the conclusion part. Need to revise and incorporate this important concern of reviewer.
- The conclusion section seems like abstract so there is a need to revise the conclusion part accordingly.
Figures and Tables:
Authors should revise the manuscript carefully in light of below comments...............
· Please provide the clear figures and tables.
· The authors should write the descriptive, elaborated legends for the figures and the tables.
· Please remove the redundancy from the legends of the figures and tables.
· The legends of the figures and tables are not crisp and not completely bringing out the sense of the figures and tables. Rewrite it accordingly.
· The placement of tables and figures in the manuscript should be done appropriately, which is missing in this manuscript. Please revise it.
· The figures are overlapping the legends, the editing needs to be done.
· The proper explanation of statistical analysis and its importance for describing the results should be mentioned.
· There should not be monotony in representation of the results for instance all should not be represented in bar graph form vise-versa.
Author Response
Response letter to editor and reviewers
Note: The line number mentioned by the editor in the comments is kept same, according to the manuscript, whereas the line number mentioned in the response for the corresponding comments line is the new line number of the modified version of the manuscript.
Reviewer 1
- Grammatical errors are present, please revise the whole manuscript to remove any possible grammatical and typos errors.
Thank you for your suggestion. The authors carefully reread the whole manuscript to revise any possible grammatical and typos errors.
- Error in sentence formation, please revise the whole manuscript to avoid the use of long sentences and confusing sentences/paragraphs.
Thank you for your suggestion. The whole manuscript was revised accordingly.
- Please maintain uniformity while in-text citation and referencing in the entire manuscript.
Uniformity was maintained in-text citation and referencing throughout the manuscript.
- The reference does not meet the format requirements of the Journal so please check the references as per the authors guideline of the Journal.
References are revised based on the journal’s reference format requirement.
- It is advised to check and avoid too many self-cited papers.
Thank you for your suggestion. We added some self-cited articles because the current research relates to our previous published articles.
- The beginning of a new paragraph should be after some space, check in complete manuscript.
Thank you for your observation. All the paragraphs in the revised manuscript begin after spaces.
- Throughout the whole manuscript the plant names should be in italic format.
Thank you for your remark. The scientific name is written in italic wherever it appears in the text.
- This paper lacks final revision by the author as many general repetitions, typos, grammatical, sentence formation errors were found in the manuscript. It is not possible to mention all such errors. Thus revise the manuscript accordingly.
Thank you for your suggestion. The entire manuscript is cross-checked and revised as suggested.
Abstract:
Authors should revise the manuscript carefully in light of below comments...............
- An abstract must be fully self-contained and make sense by itself, without further reference to outside sources or to the actual paper. It is important to provide the relevance or importance of your work and the main outcomes. Please revise the abstract accordingly.
Thank you for your suggestion. The abstract is revised accordingly.
- The abstract is not clear and the objective of the paper is not clearly validated from the abstract.
The objectives of the study are rewritten in the abstract for readers’ better understanding.
- The future perspective of the experiment should be mentioned in the abstract.
Thank you for your suggestion. The future perspective of the research is mentioned in the abstract as recommended.
- The abstract should appropriately over the contents of the manuscript.
After improvement, the authors believe the abstract appropriately covers the manuscript's contents as required.
- In the keywords, it is strongly advisable to use suitable words that can aid in finding out the manuscript in current registers or indexes. Strictly avoid the use of title words in the keywords.
Thank you for your suggestion. Keywords are revised. Although some keywords are the same as some words in the title, they are written differently.
- A graphical abstract is recommended for better perception of the present study.
Thank you for your recommendation. A graphical abstract is drawn for this study.
- A novelty statement is also encouraged to be added in the manuscript for bringing out the uniqueness of your study and its importance.
A novelty statement is added in the abstract of the revised manuscript. Lines 18-25
- Please ensure that “Highlights” of the present work should be clearly mentioned in the manuscript.
The abstract of the revised manuscript presents the “Highlights” of the study.
Introduction:
Authors should revise the manuscript carefully in light of below comments...............
- The literature from past work done in the same field missing to strengthen the introduction section. The need and importance of the present work should be clearly written in the introduction section.
Thank you for your remarks. Please note that we cited 44 references in our introduction. All the cited literature in the introduction is related to our research because they treated subjects, including salinity, melatonin treatment under salt stress, and cotton. The introduction is already over seven hundred words; adding more literature could make it much longer. However, we carefully revised and improved the introduction section.
- The new aspects and innovations of this manuscript should be clearly and briefly described in this section.
Thank you for your suggestion. The novelty statement of the study has been improved in the introduction. Lines 126-128
- The present state of knowledge in the subject should be described in introduction.
Please find the present state of knowledge in the introduction just above the novelty statement. 123-128
- The literature should be sufficiently critical, current, and internationally evaluated.
As we previously mentioned, we used 44 references related to our study in the introduction, and we are wondering if adding more literature to the length of the introduction could exceed the standard.
Materials and Methods:
Authors should revise the manuscript carefully in light of below comments...............
- Please try to merge the different sub-sections of the methodology as an individual mention for each component seems a bit unscientific method.
Thank you for your suggestion. The different sub-sections of the methodology are merged as suggested.
- The size of manuscript seems to be large. It should be crisp and appropriate. Please revise it.
The large size of the manuscript is due to the abundance of data presented in our results. We believe that each piece of data presented in our manuscript is important. That’s why they were not submitted as supplemental materials. We believe the different revisions made throughout the manuscript could give it an appropriate length.
- The text presented across the manuscript should be simple so that the scientist/workers in other disciplines will understand. Please revise it.
Thank you for your suggestion. The manuscript has been revised accordingly.
- The different sections of manuscript are poorly cited with the references and required to update and validation with previous studies. The relevant papers listed below may be considered to enhance the scientific quality of manuscript significantly.
- Mariyam, S. et al. (2023). Review on nitric oxide at the forefront of rapid systemic signaling in mitigation of salinity stress in plants: crosstalk with calcium and hydrogen peroxide. Plant Science, 111835. https://doi.org/10.1016/j.plantsci.2023.111835
- Agnihotri, A. et al. (2018). Counteractive mechanism (s) of salicylic acid in response to lead toxicity in Brassica juncea (L.) Czern. cv. Varuna. Planta, 248, pp.49-68. https://doi.org/10.1007/s00425-018-2867-0
- Prajapati, P. et al. (2023). Nitric oxide mediated regulation of ascorbate-glutathione pathway alleviates mitotic aberrations and DNA damage in Allium cepa L. under salinity stress. International Journal of Phytoremediation, 25(4), pp.403-414. https://doi.org/10.1080/15226514.2022.2086215
Thank you for your recommendation. About 78 relevant references are cited in our revised manuscript. Your suggested papers were used to improve the citation of our manuscript. Please find your suggested references in the reference list.
Results and Discussion:
Authors should revise the manuscript carefully in light of below comments...............
- The results and discussion section needs to be elaborated more. The results should be clearly described in light of available knowledge and hypothesis and must be strongly validated with previous reports in the related subject area.
Thank you for your suggestion. Authors cross-checked the whole results and discussion part to describe the results better and improve their discussions.
- The non-significant results was not clearly validated from the previous papers.
Thank you for your remark. Results and discussion are revised to create more harmony between the results and discussions.
- Please carefully check, verify, and correct the results of the present experiments from the tables/figures/graphs provided in the manuscript.
Thank you for your suggestion. All tables and figures are carefully checked and revised accordingly.
- The discussion does not describe the results with proper facts and even does not validate the result with appropriate references. Please enrich it significantly.
Thank you for your suggestion. Improvements were made to the discussion accordingly.
- The discussion did not provide a specific reasons for the results. The provided explanation should be strengthen significantly.
Thank you for your remark. In the discussion section, wherever results are not in-depth compared with previous research, specific reasons were given to explain the facts.
- The strong hypothesis, scientific facts, and validation of previous reports are entirely missing. Please revise it.
Thank you for your observation. The authors tried to improve the manuscript based on your valuable comments and suggestions. We believe our revision has reached your satisfaction level.
Conclusion:
Authors should revise the manuscript carefully in light of below comments...............
- The conclusion section failed to enlighten the spirit of the present finding or work so required to revise it accordingly.
Thank you for your suggestion. It is revised accordingly.
- In the conclusion section the authors have only mentioned the data but major finding is missing from the conclusion part. Need to revise and incorporate this important concern of reviewer.
Thank you for your suggestion. It is revised accordingly.
- The conclusion section seems like abstract so there is a need to revise the conclusion part accordingly.
Thank you for your suggestion. It is revised accordingly.
Figures and Tables:
Authors should revise the manuscript carefully in light of below comments...............
- Please provide the clear figures and tables.
All Figures and Tables are improved as suggested.
- The authors should write the descriptive, elaborated legends for the figures and the tables.
Legends are elaborated for the Figures and the Tables accordingly.
- Please remove the redundancy from the legends of the figures and tables.
Redundancy is removed from the legends of Figures and the Tables.
- The legends of the figures and tables are not crisp and not completely bringing out the sense of the figures and tables. Rewrite it accordingly.
Legends of the Figures and the Tables are revised.
- The placement of tables and figures in the manuscript should be done appropriately, which is missing in this manuscript. Please revise it.
All Figures and Tables are placed in the text next to the first position they are cited in the text.
- The figures are overlapping the legends, the editing needs to be done.\
Thank you for your remark. We revised this issue.
- The proper explanation of statistical analysis and its importance for describing the results should be mentioned.
Each Figures’ caption includes statistical analysis.
- There should not be monotony in representation of the results for instance all should not be represented in bar graph form vise-versa.
Thank you for your suggestion. Please note that each Figure represented in the bar plots is more suitable for the bar plot. Some Figures are not bar plots, and the results also include some Tables.

Reviewer 2 Report
Comments and Suggestions for Authors
The work was well executed and the results are quite interesting.
The ms requires a fair amount of editing. I have provided corrections in the annotated PDF file. All parts highlighted in yellow should be edited. For example: “combined stress” is perhaps better than “coupled stress”; the expression “MT-untreated saline” is not clear; use another expression such as “saline treatment in the absence of MT”. Also, “cotton” and “cotton seedlings” is repeated too many times in the text and this is unnecessary; use the abbreviation MT for melatonin consistently throughout the ms.
Figure 7: change “protective enzyme” to “antioxidant enzymes” and “membrane damage” to “lipid peroxidation (or “MDA content”). “Ion” is too vague.
As regards the scientific aspects:
Abstract: why is the Na+ content on the root surface equivalent to Na+ uptake? I would imagine that if the Na is taken up it is inside the root tissues, not on the surface.
Abstract and Par. 2.3. - membrane damage was not measured directly. You evaluated the extent of lipid peroxidation by measuring MDA production (which may or may not correlate with membrane damage). Avoid using the expression “membrane damage” when referring to MDA content/lipid peroxidation (also in the Discussion lines 361, 366, etc.).
Results - “Insignificantly affected” is meaningless. A treatment either affected (significantly) or didn’t. Change “insignificantly affected” to “did not significantly affect”.
Par. 2.4 - What you measured is SOD, CAT and POD activity not content. Make the necessary changes where indicated (also in the Discussion). Always refer to them as antioxidant enzymes and not “protective” enzymes.
Par. 3.5 and Fig. 5 – It is also quite confusing when you mention “Na uptake-root surface” (Fig. 4) and “Na+ absorption on the surface” (Par. 3.5). Uptake and absorption imply entry into root tissues, otherwise it is adsorption. Please expand on this aspect and explain exactly what is meant by “Na+ uptake at the seedlings’ root surface” (line 501) and what is it's physiological relevance.
Lines 321-322: what do you mean by biofilm? I assume you mean cellular membranes.
Conclusions: the conclusive remarks (lines 520-523) are not clear. Do not repeat what was done in this work but propose clear and explicit objectives for further studies on this topic.

Minor editing is required. See suggestions in the annotated PDF file.
Author Response
Response letter to editor and reviewers
Note: The line number mentioned by the editor in the comments is kept same, according to the manuscript, whereas the line number mentioned in the response for the corresponding comments line is the new line number of the modified version of the manuscript.
Reviewer 2
The work was well executed and the results are quite interesting.
The ms requires a fair amount of editing. I have provided corrections in the annotated PDF file. All parts highlighted in yellow should be edited. For example: “combined stress” is perhaps better than “coupled stress”; the expression “MT-untreated saline” is not clear; use another expression such as “saline treatment in the absence of MT”. Also, “cotton” and “cotton seedlings” is repeated too many times in the text and this is unnecessary; use the abbreviation MT for melatonin consistently throughout the ms.
Thank you for your suggestion. We have made the necessary modifications as requested.
Figure 7: change “protective enzyme” to “antioxidant enzymes” and “membrane damage” to “lipid peroxidation (or “MDA content”). “Ion” is too vague.
Thank you for your suggestion. We changed “protective enzyme” to “antioxidant enzymes” and “membrane damage” to “lipid peroxidation as suggested.
As regards the scientific aspects:
Abstract: why is the Na+ content on the root surface equivalent to Na+ uptake? I would imagine that if the Na is taken up it is inside the root tissues, not on the surface.
Thanks for your remark. The linguistic expression was incorrect. It was revised accordingly.
Abstract and Par. 2.3. - membrane damage was not measured directly. You evaluated the extent of lipid peroxidation by measuring MDA production (which may or may not correlate with membrane damage). Avoid using the expression “membrane damage” when referring to MDA content/lipid peroxidation (also in the Discussion lines 361, 366, etc.).
Thank you for your suggestion. We have made the necessary modifications as recommended.
Results - “Insignificantly affected” is meaningless. A treatment either affected (significantly) or didn’t. Change “insignificantly affected” to “did not significantly affect”.
Thank you for your suggestion. We have made the necessary modifications as recommended.
Par. 2.4 - What you measured is SOD, CAT and POD activity not content. Make the necessary changes where indicated (also in the Discussion). Always refer to them as antioxidant enzymes and not “protective” enzymes.
Thank you for your suggestion. We have made the necessary modifications as recommended.
Par. 3.5 and Fig. 5 – It is also quite confusing when you mention “Na uptake-root surface” (Fig. 4) and “Na+ absorption on the surface” (Par. 3.5). Uptake and absorption imply entry into root tissues, otherwise it is adsorption. Please expand on this aspect and explain exactly what is meant by “Na+ uptake at the seedlings’ root surface” (line 501) and what is it's physiological relevance.
Thank you for your observation. The authors agree that “surface” is inappropriate at those positions. It was deleted to keep the statements more appropriate.
Lines 321-322: what do you mean by biofilm? I assume you mean cellular membranes cellular membranes.
Thank you for your question. Exactly by “biofilm” we mean “cellular membranes”. It was revised accordingly. Lines 388-389
Conclusions: the conclusive remarks (lines 520-523) are not clear. Do not repeat what was done in this work but propose clear and explicit objectives for further studies on this topic.
Thank you for your suggestion. The conclusion was revised as recommended.
Minor editing is required. See suggestions in the annotated PDF file.
Thank you for your suggestion. They are incorporated in the revised manuscript.

Reviewer 3 Report
Comments and Suggestions for Authors
Review-Manuscript ID: plants-2640045 Foliar application of melatonin positively affects the
physio-biochemical characteristics of cotton (Gossypium hirsutum L.) under
the combined effects of low temperature and salinity stress
Authors: Yuanyuan Fu, Lang Xin, Mounkaila Hamani Abdoul Kader, Weihao Sun, Hongbo Wang, Sunusi Amin Abubakar, Xingpeng Wang, Anzhen Qin, Yang Gao
It should be noted that the authors of the manuscript have conducted a certain amount of experimental research,
However, I have reason to recommend that the authors revise the manuscript before submitting it to the journal for review.
The reference to the work in the authors' research (Lines 545-546) is not justified:4.Sun, B.; Liu, G.-L.; Phan, T.T.; Yang, L.-T.; Li, Y.-R.; Xing, Y.-X. Effects of cold stress on root growth and physiological metabolisms in seedlings of different sugarcane varieties. Sugar Tech 2017, 19, 165-175.
Evidently, there is a mistake in the reference to the work 14, (Lines 62-64) Lichtenthaler, 62 et al. [14] found that the photosynthetic functions and physicochemical properties of 63 thylakoid membranes are directly or indirectly affected by temperature and salt stresses, in which the authors of the cited article write: “Bean plants (Phaseolus vulgaris L.) were grown in the botanical garden and sugar beet plants (Beta vulgaris L.) on an experimental research field in Durlach without supply (0 kg) and with additional nitrogen fertilizer (150 kg per hectare)”.
Further, the authors write:( lines 65-68) “Melatonin (N-acetyl-5-methoxytryptamine), which was initially identified and isolated from the pineal glands of cows, is an important multifunctional hormone that is involved in a range of animal’s physiological processes modulation, such as sleep [15], immunity, and reproduction (16,17), lines 568-570, works cited: 15. Zhao, C.; Nawaz, G.; Cao, Q.; Xu, T. Melatonin is a potential target for improving horticultural crop resistance to abiotic stress. Scientia Horticulturae 2022, 291, 110560.
16. Arnao, M.B.; Hernández-Ruiz, J. The physiological function of melatonin in plants. Plant signaling & behavior 2006, 1, 89-95.
Lines: 79-81, the authors argue:” Exogenous foliar application of melatonin is involved in numerous physiological processes to improve plant resistance to salinity stress” [36]., citing the article: [36] 36. Zhao, C.; Zhang, H.; Song, C.; Zhu, J.-K.; Shabala, S. Mechanisms of plant responses and adaptation to soil salinity. The innovation 612 2020, 1, 100017, but this work discusses the mechanisms of plant resistance to salt stress and does not mention the role of foliar application of melatonin.
The methods section does not indicate on which works the choice of temperatures for the studied plants is based.
Fig 7 looks too simple, does not contain essential information, and should be removed or moved to an appendix to the article: Fig 7. “Effects of exogenous MT on the physiological mechanism of cotton seedlings under (a) salt 408 stress and (b) under coupled low temperature and salt stress”.
The conclusion should be radically rewritten.
Comments on the Quality of English LanguageMinor editing of the English language required
Author Response
Response letter to editor and reviewers
Note: The line number mentioned by the editor in the comments is kept same, according to the manuscript, whereas the line number mentioned in the response for the corresponding comments line is the new line number of the modified version of the manuscript.
Reviewer 3
Review-Manuscript ID: plants-2640045 Foliar application of melatonin positively affects the
physio-biochemical characteristics of cotton (Gossypium hirsutum L.) under
the combined effects of low temperature and salinity stress
Authors: Yuanyuan Fu, Lang Xin, Mounkaila Hamani Abdoul Kader, Weihao Sun, Hongbo Wang, Sunusi Amin Abubakar, Xingpeng Wang, Anzhen Qin, Yang Gao
It should be noted that the authors of the manuscript have conducted a certain amount of experimental research,
However, I have reason to recommend that the authors revise the manuscript before submitting it to the journal for review.
The reference to the work in the authors' research (Lines 545-546) is not justified:4.Sun, B.; Liu, G.-L.; Phan, T.T.; Yang, L.-T.; Li, Y.-R.; Xing, Y.-X. Effects of cold stress on root growth and physiological metabolisms in seedlings of different sugarcane varieties. Sugar Tech 2017, 19, 165-175.
Thank you for your remark. The Unjustified reference (4.Sun, B.; Liu, G.-L.; Phan, T.T.; Yang, L.-T.; Li, Y.-R.; Xing, Y.-X. Effects of cold stress on root growth and physiological metabolisms in seedlings of different sugarcane varieties. Sugar Tech 2017, 19, 165-175.) has been removed from the said sentence.
Evidently, there is a mistake in the reference to the work 14, (Lines 62-64) Lichtenthaler, 62 et al. [14] found that the photosynthetic functions and physicochemical properties of 63 thylakoid membranes are directly or indirectly affected by temperature and salt stresses, in which the authors of the cited article write: “Bean plants (Phaseolus vulgaris L.) were grown in the botanical garden and sugar beet plants (Beta vulgaris L.) on an experimental research field in Durlach without supply (0 kg) and with additional nitrogen fertilizer (150 kg per hectare)”.
Thank you for your observation. To avoid any mistake or incoherence in the citations, the authors deleted the statement “Lichtenthaler, 62 et al. [14] found that the photosynthetic functions and physicochemical properties of 63 thylakoid membranes are directly or indirectly affected by temperature and salt stresses”.
Further, the authors write:( lines 65-68) “Melatonin (N-acetyl-5-methoxytryptamine), which was initially identified and isolated from the pineal glands of cows, is an important multifunctional hormone that is involved in a range of animal’s physiological processes modulation, such as sleep [15], immunity, and reproduction (16,17), lines 568-570, works cited: 15. Zhao, C.; Nawaz, G.; Cao, Q.; Xu, T. Melatonin is a potential target for improving horticultural crop resistance to abiotic stress. Scientia Horticulturae 2022, 291, 110560. 16. Arnao, M.B.; Hernández-Ruiz, J. The physiological function of melatonin in plants. Plant signaling & behavior 2006, 1, 89-95.
Thank you for your observation. That section was revised and recited with more appropriate references. Lines 81-85
Lines: 79-81, the authors argue:” Exogenous foliar application of melatonin is involved in numerous physiological processes to improve plant resistance to salinity stress” [36]., citing the article: [36] 36. Zhao, C.; Zhang, H.; Song, C.; Zhu, J.-K.; Shabala, S. Mechanisms of plant responses and adaptation to soil salinity. The innovation 612 2020, 1, 100017, but this work discusses the mechanisms of plant resistance to salt stress and does not mention the role of foliar application of melatonin.
Thank you for your remark. The reference has been replaced by the accurate one. Lines 97-99
The methods section does not indicate on which works the choice of temperatures for the studied plants is based.
Thank you for your observation. Several related papers were explored before designing this experiment. Previously published articles investigated the effects of cold stress on cotton with different low temperature levels, including 4, 10, 15, and 20ºC. According to their findings, we designed our experiment with a low temperature of 15ºC, but not based on a specific paper because we conducted the experiment to make an innovation. Please find below a few articles that can be taken as reference.
- Cheng, G., Zhang, L., Wang, H., Lu, J., Wei, H., & Yu, S. (2020). Transcriptomic profiling of young cotyledons response to chilling stress in two contrasting cotton (Gossypium hirsutum L.) genotypes at the seedling stage. International Journal of Molecular Sciences, 21(14), 5095.
- Kargiotidou, A., Kappas, I., Tsaftaris, A., Galanopoulou, D., & Farmaki, T. (2010). Cold acclimation and low temperature resistance in cotton: Gossypium hirsutum phospholipase Dα isoforms are differentially regulated by temperature and light. Journal of experimental botany, 61(11), 2991-3002.
- Li, Z. B., Zeng, X. Y., Xu, J. W., Zhao, R. H., & Wei, Y. N. (2019). Transcriptomic profiling of cotton Gossypium hirsutum challenged with low-temperature gradients stress. Scientific data, 6(1), 197.
Fig 7 looks too simple, does not contain essential information, and should be removed or moved to an appendix to the article: Fig 7. “Effects of exogenous MT on the physiological mechanism of cotton seedlings under (a) salt 408 stress and (b) under coupled low temperature and salt stress”.
Thank you for your suggestion. Instead of moving Figure 7 to appendix, it has been revised and kept in the main text.
The conclusion should be radically rewritten.
The conclusion has been re written accordingly.
Minor editing of the English language required
The English has been edited and improved throughout the manuscript.

Round 2
Reviewer 1 Report
Comments and Suggestions for Authors
ACCEPT